# Higher Cognitive Reserve Is Associated with Better Working Memory Performance and Working-Memory-Related P300 Modulation

**DOI:** 10.3390/brainsci11030308

**Published:** 2021-03-01

**Authors:** Gabriela Gutiérrez-Zamora Velasco, Thalía Fernández, Juan Silva-Pereyra, Vicenta Reynoso-Alcántara, Susana A. Castro-Chavira

**Affiliations:** 1Doctorado en Investigaciones Cerebrales, Centro de Investigaciones Cerebrales, Universidad Veracruzana, Xalapa 91190, Mexico; gabrielagutierrez.zv@gmail.com; 2Laboratorio de Psicofisiología, Instituto de Neurobiología, Universidad Nacional Autónoma de México, Juriquilla, Querétaro 76230, Mexico; thaliafh@yahoo.com.mx (T.F.); castrochavirasa@inb.unam.mx (S.A.C.-C.); 3Proyecto de Neurociencias, Facultad de Estudios Superiores Iztacala, Universidad Nacional Autónoma de México, Estado de México 54090, Mexico; 4Facultad de Psicología, Universidad Veracruzana, Xalapa 91097, Mexico; maripostita@gmail.com; 5Department of Psychology, UiT the Arctic University of Norway, 09037 Tromsø, Norway

**Keywords:** cognitive reserve, sentence reading performance, P300 component, working memory, Sternberg task, reading syntactic skills

## Abstract

This study aims to examine how two levels of cognitive reserve, as evidenced by reading syntactic skill, modify performance and neural activity in a two-load-level (high vs. low) working memory (WM) task. Two groups of participants with different reading skills, high and low, were obtained from clustering analysis. We collected the P300 event-related potential component during the performance of the WM Sternberg task. The high reading performance (HRP) group showed a higher percentage of correct answers than the low reading performance (LRP) group in the negative probes of the WM task, which were probe stimuli not included in the memory set presented immediately before. Both groups showed P300 amplitude modulations, that is, larger WM-related P300 amplitudes for low than for high WM loads. Following the behavioral results, the HRP group displayed smaller WM-related amplitude modulations than the LRP group in the negative probes. The findings together suggest that higher levels of reading skill are associated with improved neural efficiency, which reflects in a better working memory performance.

## 1. Introduction

An enriched environment during one’s life may play a protective role against cognitive deficits associated with aging or pathology [1]. Such a protective effect could be a consequence of optimizing the use of resources through the recruitment of neural networks and/or alternative cognitive strategies [2]. This adaptive mechanism is called cognitive reserve (CR) [1,2]. Although CR cannot be directly measured, it has been assessed by measuring proxy variables such as educational level, intelligence quotient (IQ), occupational complexity, leisure activities that require some intellectual effort, vocabulary span [2], and reading test scores [3].

In general, reading is an activity that stimulates intellectual activity and allows the enrichment of life experiences. Reading may improve verbal comprehension and increase vocabulary, information, and general knowledge of the world [4]. Higher CR is associated with better performance in tasks evaluating working memory (WM) [5,6], verbal comprehension [4], attention, and reasoning [7,8]. Thus, CR may increase the efficiency and flexibility of neural networks by promoting brain plasticity. Additionally, high reading ability is associated with higher WM performance [9], and high text exposure with more efficient spelling and lexical processing, better attention to clauses, and better recall of sentences [4]. 

Studies with event-related potentials (ERPs) recorded during cognitive tasks allow collecting information with a high temporal resolution, which enables the exploration of the sequence of cognitive processes involved in task performance. ERPs are the average of the cerebral electrical activity synchronized to external or internal events and are classified according to their polarity (i.e., positive or negative deflections in the waves), latency (i.e., the time in milliseconds from the stimulus presentation), and distribution over the scalp [10]. In specific, the ERP P300 component is a positive deflection of maximum amplitude at approximately 300 milliseconds localized in parietal regions. [11,12].

Recently, Gu et al. [7], Speer and Soldan [6], and Sundgren, Wahlin, Maurex, and Brismar [13] showed that participants with a higher CR level had a higher percentage of correct answers, shorter response times, and a reduced WM-related modulation, the difference in P300 ERP amplitudes between high and low WM loads, than subjects with lower CR. The P300 component is associated with the updating of the WM and may be modified by the cognitive demand of a task [11,12,14] and its associated attentional processes [15]. For instance, Speer and Soldan [6] found decreased P300 amplitudes and longer latencies with the increase of WM load during a verbal Sternberg task. This WM-related modulation was higher in subjects with low CR than in subjects with high CR, which suggests that higher CR is associated with greater neuronal efficiency in terms of lower neural activity and higher processing speed as the task demand increases. Similarly, Gu et al. [7] found a negative correlation between WM-related modulation of the P300 amplitude and the level of CR; this suggests that higher CR reduces neural inefficiency. Lower modulation of the P300 amplitude is also associated with a more physically active lifestyle [16], higher educational level [17], and higher intelligence scores [18,19].

The present study aimed to examine the association between sentence-reading performance and WM as assessed by behavioral and ERP measurements, given the evidence that adults with low CR [6] show an increased WM-related P300 amplitude modulation and poor behavioral performance on a verbal WM task. Using a Sternberg task with two levels of WM load, we expect to observe a reduced percentage of correct answers and longer reaction times in a group with low reading performance (LRP) compared to a group with high reading performance (HRP). The differences between groups regarding behavioral patterns would be more evident in the high WM-load than in the low WM-load condition. Likewise, the LRP group would show a significantly greater WM-related modulation with smaller P300 amplitudes as the WM load increases, while the HRP group would not show this pattern of amplitude modulation because these conditions would not have a significantly higher processing cost for them.

## 2. Materials and Methods

### 2.1. Participants

The sample consisted of 39 young Mexican adults, 20 women, and 19 men. All participants were right-handed, and their ages ranged between 20 and 30 years (mean age = 24.46 years, SD = 3.04). All participants had a normal or corrected-to-normal vision and did not report any history of neurological or psychiatric problems, illegal drug use, or frequent consumption of medical drugs or alcohol. All participants provided informed consent. The project was approved by the Ethics Committee of the Institute of Neurobiology of the Universidad Nacional Autónoma de México (Ethical Application Ref: INEU/SA/CB/109), following the Declaration of Helsinki.

A hierarchical cluster analysis was applied to identify possible homogeneous subgroups according to a grammatical judgment reading task previously used elsewhere [20]. Only sentences that included a clause between the main noun and the adjective were considered in this study. The sentences varied in the gender agreement (agree or disagree) between noun and adjective (La casa que está en la colina es amarilla [The house (feminine) over the hill is yellow (feminine)]; El edificio que está en la colina es amarilla [The building (masculine) over the hill is yellow (feminine)]). Participants had to read the whole sentence (40 agreement and 40 disagreement sentences) and respond as quickly and accurately as possible only when a question mark appeared. Participants judged the grammaticality of the sentences by pressing a button on a response box. One button was for ‘‘correct” sentences (agreement sentences), and the other button was for ‘‘incorrect” sentences (disagreement sentences).

Standardized Z scores for the numbers of correct answers to sentences, for both gender agreement and gender disagreement categories, were used to perform cluster analysis. The Ward method employed incorporated a measure of squared Euclidean distance since this distance is sensitive to the variable metrics. Visual inspection of the dendrogram revealed two independent clusters that were similar in size. Two groups were obtained: HRP n = 18 (10 females) and LRP n = 21 (10 females). There were no significant differences between the groups regarding gender proportion (X^2^ = 1.40, p = 0.32). Table 1 summarizes the sociodemographic features and the sentence-reading behavioral performance of both groups.

### 2.2. Sternberg Task

Stimuli consisted of 180 digit-sets of two levels of memory load. The memory sets contained four digits (1 to 9). Every low-load memory set consisted of a digit repeated four times, while the high-load memory sets consisted of four different digits randomly sorted, avoiding strings of consecutive numbers either in ascending or descending order. The digits were presented in white characters (Arial 12) with a black background. The probe stimuli consisted of one digit placed at the center of the monitor.

Stimuli were delivered by the STIM2 software (NeuroScan, CompuMedics, Charlotte, NC, USA) through a PC using a 17” monitor. Participants sat 70 cm away from the screen in a Faradized sound-attenuatinganddimly-lit room.

Each trial began with the presentation of an asterisk (fixation point) at the center of the screen for 1000 ms, followed by an interstimulus interval of 500 ms. Subsequently, a memory set presented for 1000 ms was followed by a black screen (1000 ms; memory retention period), as shown in Figure 1. Finally, a probe digit appeared for 1000 ms, and the participants responded by pressing either one button if the probe digit appeared in the preceding memory set (positive probe) or the other button if the probe digit was not present (negative probe). The counterbalance of button usage extended across the sample. Trials formed 45-trial blocks. The trial presentation was random and the same for all subjects. The participants received the instruction to answer as quickly as they could while avoiding making mistakes. Participants did not receive feedback on their performance. The task lasted approximately 20 min.

### 2.3. ERP Acquisition and Analysis

Each electroencephalogram (EEG) was recorded using a SynAmps system (NeuroScan, CompuMedics, Charlotte, NC, USA) and collected through 32 Ag/AgCl electrodes embedded in an elastic cap (ElectroCap International, Inc., Eaton, OH, USA) according to the 10/20 international system and referenced to the right earlobe (A2). The left earlobe was recorded independently. The bandwidth of the amplifiers was set to 0.1–100 Hz, and the signal was digitized at a 500-Hz sampling rate. Impedances were kept below 5 kΩ. Two electrodes placed on the external canthus and superciliary arch of the left eye were used to record the electrooculogram.

The EEG recordings were re-referenced offline using the average of the earlobe signals (A1-A2). The continuous EEG recording was epoched from 200 ms pre-stimulus to 1000 ms post-stimulus. The ERP waveform was baseline corrected, and drift was removed using the linear detrend tool of the NeuroScan 4.5 software (CompuMedics; Charlotte, NC, USA). Only segments corresponding to correct answers were analyzed. All EEG epochs were visually inspected, and manual rejection of segments was performed. The artifact-free segments per the experimental condition of each participant were averaged. The same number of segments was used to build the ERPs’ average across the two groups and the two conditions.

### 2.4. Statistical Analysis

The nonparametric permutation test was applied to the behavioral and ERP data using 10,000 permutations. This method is a distribution-free test that builds an empirical probability distribution computing multiple statistical tests. This statistical procedure is included in the eLORETA software [21]. For ERP data analysis, the t-tests were performed for each electrode. The maximum t and the extreme probability values as well as t- and *p*-values were reported.

## 3. Results

### 3.1. Behavioral

Comparisons between groups (HRP and LRP) were performed for the four experimental conditions (low-load WM positive probes, high-load WM positive probes, low-load WM negative probes, and high-load WM negative probes) using the percentage of correct answers (CA) and the response times.

CAs for each of the experimental conditions are shown in Figure 2a. The HRP group displayed a significantly greater CA than the LRP group for the high-load WM negative probes (t max = 2.599, extreme *p* = 0.036) and this difference did not reach significance for the high-load WM positive probes (t = 2.265, *p* = 0.0698). Groups did not differ in CA for low-load WM positive (t = 0.46, *p* = 0.65) or for low-load WM negative probes (t = 1.80, *p* = 0.14).

We computed the WM-related effect CA-change subtracting high WM from low WM load per probe type. The HRP group showed significant lower WM-related effect than the LRP group for negative probes (t max = −2.10, *p* = 0.045), but not for positive probes (t = −1.92, *p* = 0.07).

Regarding time responses, there were no differences between groups, although the LRP group showed a tendency for longer response times in all conditions, as is shown in Figure 2b.

### 3.2. ERP Epochs

ERP epochs for amplitude analyses were defined as the time windows where the ERP amplitude significantly differed between high and low WM loads throughout the whole window (−200 to 1000 ms) and across all electrodes over the scalp. Analyses were separately performed per group and probe type combination (Figure 3). Significant differences between WM loads were found in the 226 ms–388 ms (HRP group) and 234–370 ms (LRP group) epoch intervals for positive probes; and in the 266–412 ms (HRP group) and 298 - 422 ms (LRP group) epoch intervals for the negative probes. Within these ERP epochs, a positive wave had a significantly larger amplitude for the low WM load than for the high WM load condition. Considering latency of occurrence, polarity, and correspondence of this component with the cognitive task, the significant effect observed can be regarded as a WM-related P300 modulation.

We obtained the topography of the modulations per combination of WM load and probe type using the mean amplitude of the epochs. The HRP group showed a significant WM-related modulation over all the electrode sites except on Fp2 for positive probes, and WM-related modulations over all the electrode sites except on Fp1, Fpz, and Fp2 for negative probes. The LRP group showed WM-related modulations over all the electrode sites except Fp1, Fpz, O1, and F7 for positive probes, and WM-related modulations over all the electrode sites for negative probes. Significance for all the analyses was set at *p* < 0.05.

### 3.3. ERP Amplitude

WM-related modulations (i.e., low WM minus high WM loads) were computed for positive and negative probes. We compared the epoch mean amplitudes between groups per probe-type at all electrode sites. The HRP group (266–412 ms) displayed a lower WM-related modulation than the LRP group (298–422 ms) over Fp2 (t max = −2.96, extreme *p* = 0.0276) and Fpz (t = −2.76, *p* = 0.046; Figure 4a). The amplitude differences per group at Fpz and Fp2 can be observed in Figure 4b, which shows significant group differences in negative probes. In contrast, no difference between groups was observed for the positive probes.

## 4. Discussion

A large body of research suggests that WM is associated with reading comprehension ability [22], but few show an association of reading syntactic skill and CR with WM.

In this study, we contrasted two levels of WM loads in participants with high (HRP) and low (LRP) sentence reading performance based on the participant’s accuracy in a syntactic judgment reading task. In specific, we addressed the issue of whether high CR, assessed as better sentence reading performance, underlies better WM performance and neural efficiency. In general, the participants with HRP were expected to show more accuracy and faster responses than those with LRP when identifying whether the probe belongs or not to a preceding memory set; in specific, the HRP group would show smaller differences in behavioral performance between WM-load levels than the LRP group. Our results partially supported our hypotheses. The participants with HRP significantly displayed more correct answers and smaller WM-related modulations than those with LRP only in the negative probes (i.e., probes not included in the memory set). In addition to having greater syntactic skills, the HRP group had significantly more years of schooling, higher WAIS-similarities scores, WAIS-information scores, and WAIS-verbal comprehension indices than the LRP group; all these scores are considered proxy measurements of CR. These findings concur with the notion of reading skills as a measure of CR [3]. Our results also support the idea that higher CR may be associated with a more efficient performance of a WM task [6,11]. Moreover, better sentence reading performance may be associated with higher CR in a positive feedback loop [4] that might favorably impact verbal WM. Most CR studies involve older adults or patients with some class of brain damage. In this study, the participants are healthy young people in their third decade of life, so it is reasonable to assume full integrity of the CR substrate, i.e., the brain reserve [23]. Despite this, the effects of brain plasticity induced by higher CR are observable as better performance in a verbal WM task.

Speer and Soldan [6] and Gu et al. [7] found that individuals with higher CR showed a reduced WM-related P300 modulation with increasing task demands, which would reflect an underlying neural efficiency [24]. Based on these findings, we expected that the participants with HRP would show smaller modulations in the P300 amplitude than those participants with LRP as the WM load increased. Thus, the P300 amplitude modulation would reflect greater neural efficiency if neural resources were allocated similarly to process both high and low WM loads [6]. However, if more neural resources were allocated to process a higher cognitive demand, then the neural response would be less efficient. Our ERP results support this hypothesis, but only for the negative probe condition.

The negative probe condition is likely to pose higher cognitive demands than the positive probe because there must be an exhaustive search for the probe within the memorized set; besides, the subject has to figure out that the probe does not belong to the memory set, and probably discard useless information to answer correctly. This idea concurs with the fact that poor comprehenders have a specific difficulty suppressing irrelevant information when there is a high number of items to suppress [25]. This difficulty in avoiding intrusions is not due to an increased memory load or retrieval effort but is related to the quantity of irrelevant information that the subject needs to suppress. Evidence suggests that the P300 response with fronto-central localization may be a marker of inhibition [26].

Previous studies have shown that subjects who remembered fewer stimuli displayed more activity in the bilateral frontal cortex. The recruitment of these frontal regions is also associated with less education [27]. Accordingly, our participants with LRP, which presented less neural efficiency than those with HRP, showed this effect over frontal sites. That is, the involvement of frontal regions is superior in participants with LRP who strive to evaluate whether the probe stimulus belongs to the memory set, perhaps due to having more difficulty in suppressing irrelevant information. Thus, our data suggest that participants with LRP who are challenged by a cognitive task, and consequently perform worse, engage the frontal cortex to a greater degree. A similar relation between increased frontal activity and less education has been found elsewhere [27] and indicates that individuals with less education also tend to be those who expend more effort on cognitive tasks. Evidence provided by Franzmeier et al. [28] suggests that individuals with higher CR present higher efficiency of networks that involve the frontal cortex and regions associated with memory tasks.

CR may improve to the extent the lateral surface of the right prefrontal lobe and the right inferior parietal lobe maintain structural (white and gray matter) and functional integrity and connectivity [29,30]. Arousal, novelty, attention, awareness, and working memory share similar neural circuitry, strongly associated with frontal activity and noradrenergic function. Noradrenergic activity may mediate the effects of environmental enrichment on neurogenesis and enhanced memory and may act as a neuromodulator, increasing synaptic plasticity.

Novel stimuli processing, measured by the P300 responses, is associated with genes that influence noradrenergic availability [31]. This association accords with strong evidence that the P300 novelty response is related to the noradrenergic activity [13,32]. Locus coeruleus (LC) is the principal source of noradrenaline (NA) in the brain. The LC-NA system is critical for the regulation of cognitive performance. According to Nieuwenhuis et al. [32], P300 reflects the LC-NA system response to the outcome of stimulus evaluation and perceptual decision making. The simultaneous occurrence of P300 and autonomic components of the orienting response reflects coactivation of the LC-NA system and the peripheral sympathetic nervous system through their common major afferent: the rostral ventrolateral medulla, a key sympathoexcitatory region [33].

## 5. Conclusions

A higher level of syntactic reading skills, reflecting higher scores in proxy measurements of cognitive reserve, seems to be associated with better behavioral performance and greater neural efficiency in working memory.

## Figures and Tables

**Figure 1 brainsci-11-00308-f001:**
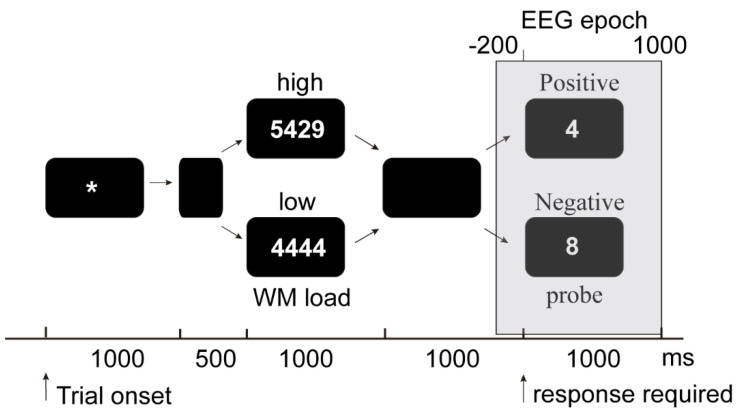
Flowchart of the Sternberg working memory (WM) task. Examples of memory sets and probes for the high and low WM load conditions that either do or do not belong to the set (positive and negative probes, respectively). The gray box represents a time-locked window of the continuous EEG.

**Figure 2 brainsci-11-00308-f002:**
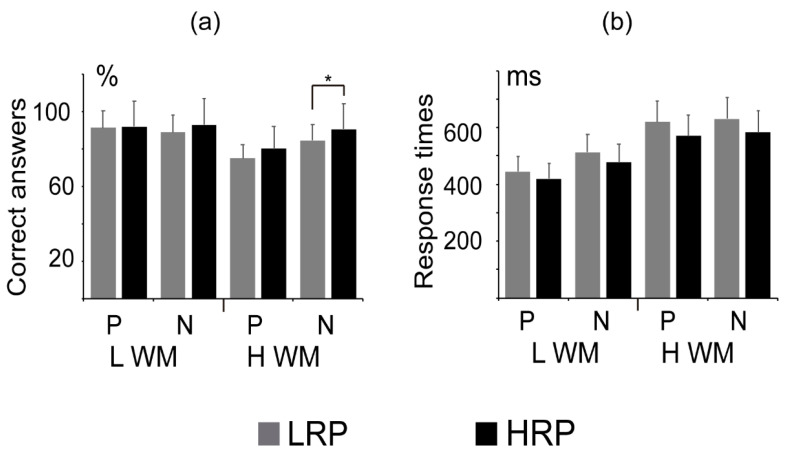
Means and standard deviations of the percentage of: (**a**) correct answers and (**b**) response times, for high- and low-load working memory sets (L WM and H WM, respectively) for positive (P) and negative (N) probes. Note that both groups displayed significantly lower percentages of correct answers for the H WM than for the L WM condition. This effect was larger for the group with low reading performance (LRP, represented in gray bars) than for the high reading performance group (HRP, black bars); * *p* < 0.05.

**Figure 3 brainsci-11-00308-f003:**
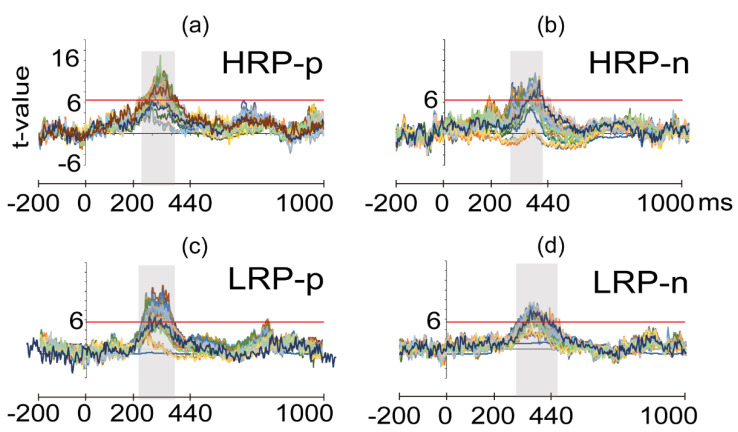
Event-related potentials of the Sternberg task. t-values across epochs (–200 to 1000 ms) of every electrode site (in various colors) resulting from the comparison between low and high WM loads for: (**a**) high reading performance group in the positive probes (HRP-p); (**b**) high reading performance group in the negative probes (HRP-n); (**c**) low reading performance group in the positive probes (LRP-p); (**d**) low reading performance in the negative probes (LRP-n). Red horizontal lines represent the threshold of significant t-values at *p* < 0.05. Gray shadow boxes indicate the time windows for significant amplitude differences.

**Figure 4 brainsci-11-00308-f004:**
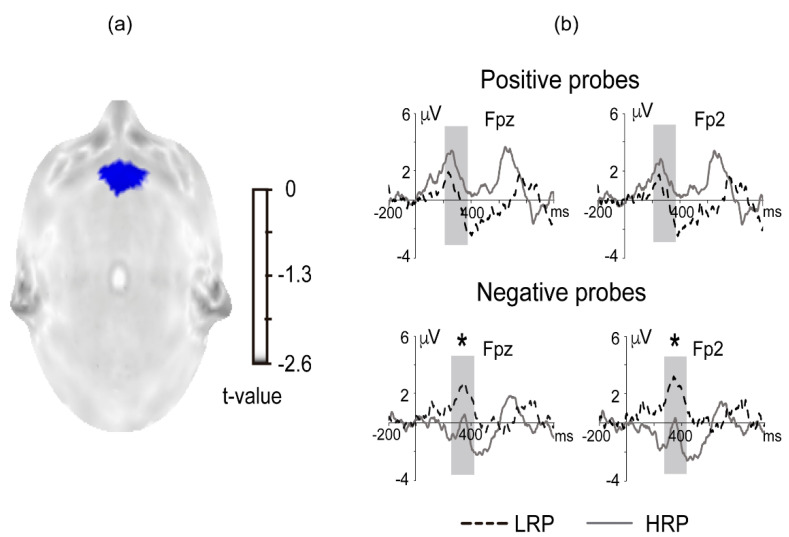
(**a**) Statistical t map of WM-related change of the P300 for the comparison between groups of negative probes. Blue represents t-values, where the P300 amplitude difference (low WM load > high WM load) was significantly smaller. (**b**) Grand average difference waves (low WM load minus high WM load) of the groups per type of probe at Fpz and Fp2. Black solid lines represent the HRP group, and dotted gray lines represent the LRP group. Gray shadow boxes represent the analysis epoch, and the asterisks indicate significant differences between groups; *p* < 0.05.

**Table 1 brainsci-11-00308-t001:** Demographic data and WAIS subscale scores. Means and standard deviations.

	LRP	HRP			
N (female)	21 (10)	18 (10)			
	Mean (SD)	Mean (SD)	t (37)	*p*	95% CI
Age (years)	24.76 (2.13)	24.11 (2.33)	0.67	0.510	−2.61:1.31
Years of Schooling	11.90 (3.36)	14.78 (3.23)	2.71	0.010	0.73:5.02
WAIS					
Similarities	9.43 (1.81)	11.17 (2.50)	2.51	0.016	0.34:3.14
Vocabulary	10.05 (2.67)	11.50 (2.26)	1.82	0.078	–0.17:3.07
Information	8.24 (3.69)	10.61 (3.24)	2.12	0.041	0.10:4.64
VCI	95.76 (12.88)	106.78 (13.91)	2.57	0.014	2.32:19.71
Reading					
Agreement	55.12 (8.82)	79.44 (7.45)	9.33	<0.001	19.05:29.61
Disagreement	54.64 (9.13)	70.42 (11.83)	4.60 ^a^	<0.001	8.79:22.75

SD: Standard Deviation; VCI: Verbal Comprehension Index; ^a^: df = 31.8.

## Data Availability

The data presented in this study are openly available in FigShare at doi:10.6084/m9.figshare.13624082.

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
