# Peer review of "Higher Cognitive Reserve Is Associated with Better Working Memory Performance and Working-Memory-Related P300 Modulation"

_brainsci, 2021, doi:10.3390/brainsci11030308_

Round 1
Reviewer 1 Report
I think this study makes an important contribution on the relationship between reading skills, as a measure close to the cognitive reserve, and neural efficiency in a working memory task. In the introduction relevant information that supports the proposed hypotheses is provided, these hypotheses are correctly raised, the research methodology is in line with the hypotheses and results are discussed properly connecting the results with the hupotheses and previous literature on this investigation area.
Author Response
Thank you very much for your comments about the paper
Reviewer 2 Report
In general it is a nice-to-read article, but I'd like to add some comments in order to improve it.
The location of the explanation of the P300 seems to be very abrupt in the middle of the paragraph. It would be better if you could explain it a little before
In Figure 3 labels "a" and "b" are in bold whereas "c" and "d" are not.
Figure 4 has the "a" and "b" labels in the previous page. I’d also label all the axes in figure b. It is also necessary to label all the graphs in order to know which electrodes they are and tell which , not only in two of the graphs
In the ERP analyses it isn’t quite clear whether the t-tests are performed individually for each electrode or if they are performed in clusters. I think it is necessary to change that.
Author Response
Reviewer 2
In general it is a nice-to-read article, but I'd like to add some comments in order to improve it.
The location of the explanation of the P300 seems to be very abrupt in the middle of the paragraph. It would be better if you could explain it a little before.
Response: We agree that the P300 definition should be introduced before in the text to improve clarity; thus, its definition was set immediately after the ERP description and just before the mentioned paragraph (lines 56,57 and 62,63).
In Figure 3 labels "a" and "b" are in bold whereas "c" and "d" are not.
Response: We apologize. We fixed that problem.
Figure 4 has the "a" and "b" labels in the previous page. I’d also label all the axes in figure b. It is also necessary to label all the graphs in order to know which electrodes they are and tell which , not only in two of the graphs.
Response: Thanks a lot for your suggestions. We modified the figures.
In the ERP analyses it isn’t quite clear whether the t-tests are performed individually for each electrode or if they are performed in clusters. I think it is necessary to change that.
Response: The t-tests were performed separately for each electrode. We added this detail to the Statistical Analysis section of the article (lines 175-176).